

# MFAM-AD: an anomaly detection model for multivariate time series using attention mechanism to fuse multi-scale features

Shengjie Xia, Wu Sun, Xiaofeng Zou, Panfeng Chen, Dan Ma, Huarong Xu, Mei Chen and Hui Li

State Key Laboratory of Public Big Data, College of Computer Science and Technology, Guizhou University, Guiyang, China

## ABSTRACT

Multivariate time series anomaly detection has garnered significant attention in fields such as IT operations, finance, medicine, and industry. However, a key challenge lies in the fact that anomaly patterns often exhibit multi-scale temporal variations, which existing detection models often fail to capture effectively. This limitation significantly impacts detection accuracy. To address this issue, we propose the MFAM-AD model, which combines the strengths of convolutional neural networks (CNNs) and bi-directional long short-term memory (Bi-LSTM). The MFAM-AD model is designed to enhance anomaly detection accuracy by seamlessly integrating temporal dependencies and multi-scale spatial features. Specifically, it utilizes parallel convolutional layers to extract features across different scales, employing an attention mechanism for optimal feature fusion. Additionally, Bi-LSTM is leveraged to capture time-dependent information, reconstruct the time series and enable accurate anomaly detection based on reconstruction errors. In contrast to existing algorithms that struggle with inadequate feature fusion or are confined to single-scale feature analysis, MFAM-AD effectively addresses the unique challenges of multivariate time series anomaly detection. Experimental results on five publicly available datasets demonstrate the superiority of the proposed model. Specifically, on the datasets SMAP, MSL, and SMD1-1, our MFAM-AD model has the second-highest F1 score after the current state-of-the-art DCdetector model. On the datasets NIPS-TS-SWAN and NIPS-TS-GECCO, the F1 scores of MAFM-AD are 0.046 (6.2%) and 0.09 (21.3%) higher than those of DCdetector, respectively(the value ranges from 0 to 1). These findings validate the MFAMAD model's efficacy in multivariate time series anomaly detection, highlighting its potential in various real-world applications.

Corresponding authors
Mei Chen, 1004225928@qq.com, gychm@qq.com
Hui Li, huili.gm@gmail.com

## INTRODUCTION

Multivariate time series anomaly detection is important in domains such as IT operations, as these series provide a direct representation of system functionality and operations, serving as a crucial foundation for the identification of anomalies. Instances of such data

encompass various monitoring indicators within operation and maintenance systems. Conventional methods, including statistical approaches and machine learning techniques, have been widely utilized in time series anomaly detection. Nevertheless, it has become increasingly apparent that conventional methodologies are inadequate when dealing with high-dimensional time series data, due to the rapid expansion of data dimensions and the associated increase in complexity. Research has shown their effectiveness has failed to meet real-world requirements (*Wu, 2016*; *Cook, Msrl & Fan, 2020*).

Concurrently, the deep neural network model has been undergoing rapid advancements, exhibiting remarkable proficiency in handling high-dimensional data, particularly those with spatiotemporal characteristics. Numerous studies have consistently demonstrated the superior performance of deep learning based anomaly detection techniques compared to traditional methods in practical applications across various time series anomaly detection techniques (*Audibert et al., 2022*). This has opened up new horizons for research and development in this domain. Because of this, using deep learning algorithms to find outliers in multivariate time series is an important area of research in machine learning (*Cook, Msrl & Fan, 2020*).

Unsupervised and supervised anomaly detection are the two fundamental strategies utilized in the realm of anomaly detection. Unsupervised methods offer practical benefits, given the difficulty associated with manually labeling real-world datasets, exacerbated by the issue of class imbalance stemming from the scarcity of anomalous data. Within the ambit of unsupervised deep learning for anomaly detection, prediction models and reconstruction models emerge as the preeminent techniques. These models differ in their conceptual frameworks and methodological approaches to the problem of anomaly detection.

Reconstruction models demonstrate resilience to anomalously sparse data and effectively learn from normal samples during the training phase. However, a significant limitation of these models lies in their often insufficient robustness when extracting temporal features. A prime example is the AutoEncoder (AE), a widely used reconstruction model algorithm that achieves data reconstruction by mapping inputs to a hidden space. In contrast, predictive models rely heavily on anomaly scores to identify outliers. These scores quantify the discrepancy between expected and observed values, offering a quantifiable metric for anomaly detection. Long short-term memory (LSTM) models stand out for their notable temporal modeling capabilities, particularly when dealing with sequential data. Despite their strengths, when confronted with long-term series data, these models may encounter issues such as the gradient explosion problem. Furthermore, dealing with data sparsity poses another challenge, often rendering the learning process unstable and challenging.

To overcome the challenges posed by complex time series data, we integrate sequence models like LSTM with reconstruction models. This innovative combination harnesses the unique temporal modeling capabilities of LSTM and the robustness of reconstruction models, leading to a significant improvement in anomaly detection accuracy. This integration not only boosts the overall performance of the model but also introduces innovative methods for addressing intricate problems. In this context, we introduce MFAM-AD (Multi-scale Features Anomaly Detection), a state-of-the-art model for multivariate time series anomaly detection. MFAM-AD builds upon the foundations

of convolutional neural network (CNN) architecture and bi-directional long short-term memory (Bi-LSTM). This model leverages the power of reconstruction to identify anomalies by comparing the reconstructed data with the original input. Significant deviations between the two indicate the presence of anomalies, which arise because the model is trained primarily on regular time series data. This training process may result in less accurate reconstructions for anomalous data exhibiting unfamiliar patterns. By reconstructing the input data, MFAM-AD pinpoints anomalies, enabling the identification of previously unrecognized errors. This practical approach not only enhances the utility of the model but also expands its capabilities in addressing real-world challenges. Overall, MFAM-AD offers a robust and effective solution for multivariate time series anomaly detection.

In our study, we introduce a novel approach for anomaly detection in multivariate time series data. Specifically, we employ a Bi-directional long short-term memory (Bi-LSTM) network to capture temporal dependencies within the data, following the utilization of a multi-scale CNN. Subsequently, the input data undergoes reconstruction *via* a decoder that incorporates a fully connected neural network (FCNN) and Bi-LSTM components. The fundamental principle of our anomaly detection method lies in the analysis of disparities between the reconstructed data and the original data. This comparison serves as the basis for anomaly determination, enabling us to detect anomalies within multivariate time series with greater reliability and precision.

The key contributions of our work are highlighted as follows:

- **Introduction of MFAM-AD:** We introduce MFAM-AD, a cutting-edge model for multivariate time series anomaly detection. This model incorporates an attention mechanism to effectively fuse multi-scale features, offering a significant improvement in recognition accuracy compared to traditional detection techniques.
- **Multi-scale feature fusion with attention mechanism:** By leveraging the attention mechanism, we successfully achieve multi-scale feature fusion, presenting a novel and effective approach for anomaly detection in time series data.
- **Rigorous evaluation:** We comprehensively evaluate the performance of our model using various evaluation metrics and publicly available datasets. This evaluation substantiates the efficacy and validity of our approach, demonstrating its superiority over existing methods.

## RELATED WORK

Anomaly detection plays an important role in improving data accuracy, identifying potential threats, enhancing security protection capabilities, and achieving widespread applicability. *Kong et al. (2024)* identifies abnormal patterns or trajectories in urban environments by detecting abnormal movements of cars and providing timely warnings to drivers to avoid accidents, highlighting the importance of anomaly detection. In our investigation, we concentrate primarily on detecting anomalies within time series data, delineating our focus into two pivotal domains: multivariate time series anomaly detection and multiscale feature time series anomaly detection. These two fields are integral to our research, providing the theoretical and empirical frameworks for our proposed approach.

## Multivariate time series anomaly detection

In general, the detection of anomalous patterns within data often involves the utilization of methodologies such as the autoregressive integrated moving average (ARIMA) model, exponential smoothing, and moving averages. These methodologies are effective in pinpointing outliers that deviate from the expected trend, leveraging the inherent patterns and statistical properties of time series data.

On the other hand, machine learning approaches leverage algorithms for clustering and classification to detect anomalies. Techniques such as density-based approaches and k-means clustering algorithms are employed to partition high-dimensional time-series data into multiple clusters, facilitating the identification of anomalies that stand out distinctly from other data points. The goal of classification algorithms such as Support Vector Machine (SVM) is to determine the hyperplane with the maximum margin in high-dimensional time series data and use this hyperplane to distinguish between normal and abnormal data points.

However, the escalating complexity of multivariate time series data poses significant challenges for the application of statistical learning and machine learning techniques. As datasets become increasingly intricate, traditional methodologies often encounter limitations in their ability to effectively capture and characterize anomalous behavior.

Deep learning techniques exhibit distinct advantages when addressing the challenge of time series anomaly detection. Models such as long short-term memory (LSTM) and autoencoder (AE) are capable of unraveling complex patterns and intrinsic correlations within data, particularly in scenarios involving high-dimensional or nonlinear data. LSTM can effectively deal with complex multivariate time series data by effectively capturing the inherent long-term dependencies and short-term patterns in sequence data.

For instance, *Hawkins et al. (2002)* established the fundamental principle of autoencoders within a multilayer perceptron neural network architecture. The method uses reconstruction error as the evaluation index and input variables as the output target to train the model, and realizes the accurate detection of outliers. In contrast, *Malhotra et al. (2016)* proposed an encoder–decoder model that employed LSTM, demonstrating exceptional effectiveness in processing time-series data generated by sensors monitoring the operational behaviors and health statuses of mechanical devices. This model adeptly pinpoints anomalies in the operation of mechanical equipment by comprehensively analyzing these data streams.

Furthermore, *Yang et al. (2023)* introduced a dual-attention contrast representation learning-based technique into time series anomaly detection, enhancing the performance of anomaly detection. This innovative method highlights the continuous progress of using deep learning methods to solve the challenge of complex anomaly detection in time series data.

Multivariate time series anomaly detection models are primarily categorized into two general classes: supervised and unsupervised anomaly detection models. Given the scarcity and expense of labeled datasets in real-world settings, unsupervised learning algorithms stand out as particularly advantageous and widely applicable across a diverse range of anomaly detection applications. The challenge of detecting anomalies within multivariate

time series data can be efficiently addressed through diverse unsupervised learning strategies, encompassing cutting-edge deep learning techniques as well as conventional classical algorithms. In the absence of prior label information, these techniques adeptly reveal the underlying structure and patterns within the data, enabling precise localization and identification of anomalous patterns. Consequently, unsupervised anomaly detection models such as InterFusion (*Li et al., 2021*), TranAD (*Tuli, Casale & Jennings, 2022*), AnomalyBERT (*Jeong et al., 2023*), and other recent proposals have gained significant popularity in contemporary multivariate time series anomaly detection endeavors.

## Multiscale feature time series anomaly detection

Anomaly detection holds significant importance in the field of multi-scale feature time series. The objective is to detect and identify abnormal patterns or behaviors that span multiple time scales. This research domain demands a rigorous examination of time series data, originating from diverse sources like financial markets, biomedical signals, and sensor data. The challenge lies in the simultaneous consideration of multiple time scales, as anomalies can manifest in various forms and scales.

The wavelet transform mode maxima method emerges as a comprehensive approach for anomaly detection across all scales. This method enables the analysis of local signal behavior, effectively pinpointing weak transient and singular features within the time series data. Moreover, mastering the extraction of pertinent information from multi-scale features and its application to anomaly detection is crucial in this field of research.

Furthermore, the utilization of advanced deep learning techniques, such as recurrent neural networks (RNNs) and CNNs, holds the potential to significantly enhance the accuracy and efficiency of multi-scale feature time-series anomaly detection. For instance, *Chen et al. (2022)* introduced an innovative hierarchical temporal context encoding block that leverages graph convolution and multi-scale extended convolution to effectively detect anomalies in multivariate time series data within the Internet of Things domain. Similarly, *Xu et al. (2018)* proposed a methodology that utilizes multi-scale time-frequency information extracted from discrete wavelet transforms, in combination with temporal convolutional networks (TCNs), to achieve fault diagnosis in industrial processes.

Recognizing the advantages of deep learning models in unsupervised anomaly detection, we aim to integrate the feature extraction capabilities of CNNs and the temporal dependency modeling abilities of LSTMs to enhance the performance of reconstructing time series data so as to improve anomaly detection. To this end, we introduce a novel time series anomaly detection model named MFAM-AD. This model employs a joint coding approach, incorporating both CNNs and Bi-LSTMs to capture both local and temporal context information from multidimensional time series data across various scales. Additionally, the encoding module incorporates an attention mechanism to effectively realize feature fusion. During the decoding stage, the model leverages FCNNs and Bi-LSTMs for data reconstruction, enabling more precise identification of anomalous behaviors within the time series. Through this innovative approach, MFAM-AD aims to provide a robust and accurate solution for multi-scale feature time-series anomaly detection.

# PROBLEM STATEMENT

In real-world contexts, the acquisition of labeled data usually requires high cost, thereby accentuating the importance of unsupervised learning technologies. Anomalies usually exhibit a series of different characteristics, from transient peaks or deviations to persistent anomaly patterns. Traditional anomaly detection methods often restrict their focus to a single time scale, limiting their ability to detect anomalous events comprehensively and precisely. Given that anomalies in time series data can manifest across multiple time scales, multi-scale time series anomaly detection aims to incorporate information from various time scales simultaneously, thereby enhancing anomaly identification. Specifically, the following challenges need to be addressed in multi-scale time series anomaly detection:

- Multi-scale feature extraction: A significant challenge in multi-scale time series anomaly detection involves devising methodologies for extracting features from time series data across varying time scales. Processing time series data involves segmenting it into different granularities, which adds complexity to the analysis.
- Feature fusion: Another pivotal aspect of multi-scale time series anomaly detection is the efficient integration of features extracted from diverse time scales to construct a comprehensive feature representation. Effective feature fusion necessitates the consideration of correlations, redundancies, and potential conflicts among different features to ensure an optimal representation.
- Design of anomaly detection algorithms: Crafting suitable anomaly detection algorithms is crucial for identifying anomalous events within time series data. These algorithms must leverage the extracted and fused features to effectively discern anomalies amidst the data.

We assess the current state of an entity by analyzing time series collected from multiple sensors, which can be expressed in the following form:

$$S = y(X_i) \tag{1}$$

where $X_i$ represents the data at the current timestamp, S represents whether it is abnormal or not, a value of 1 signifies the entity is in an abnormal state at the $i$th timestamp, and a value of 0 signifies the entity is operating normally. Our goal is to design a model that does the above and solves the problems mentioned earlier.

Firstly, suppose there is a multivariate time series $X$ from multiple sensors. We segment it using a fixed-length window w, and obtain the subsequence set x = $\{x_1, x_2, \ldots, x_n\}$. We need to design a module f that can extract features of different scales by changing its parameters $n$. For each subsequence $x_i$, we can extract a series of features $F_N$, (where $N = 1, 2, \ldots, w$). This series of features is our proposed multi-scale feature.

$$F_N = f(N, x_n). \tag{2}$$

Secondly, assume that two features, $\mathbf{F}_1$ and $\mathbf{F}_2$ have been extracted from varying time scales. To obtain novel features $K$, it is imperative to devise a fusion mechanism $c$ that effectively combines these features of different scales.

$$K = c(F_1, F_2). \tag{3}$$

Finally, recognizing the significance of temporal features in time series analysis, it is crucial to devise a module specifically aimed at capturing the temporal feature $T$. Subsequently, by integrating $T$ with the spatial feature $K$, we aim to derive the spatio-temporal feature $Z$. The feature $Z$ is then seamlessly incorporated into our anomaly detection algorithm to enhance its effectiveness.

$$Z = y_t(T, K). \tag{4}$$

By following the aforementioned steps, our model is capable of executing anomaly detection for entities, thereby addressing the previously mentioned challenges.

## THE MFAM-AD MODEL

### Overview

To effectively detect anomalies, we introduce a novel model, named MFAM-AD, that seamlessly integrates features across multiple time scales. The MFAM-AD model is composed of two fundamental components: the encoder and the decoder. Figure 1 provides a comprehensive overview of the basic architecture of the MFAM-AD model. The encoder plays a crucial role in generating hidden variables by mining intrinsic features such as temporal and multi-scale spatial features of multivariate time series data. This design enhances the model's detection accuracy, enabling MFAM-AD to effectively consider scenarios where anomalies may occur across various time scales. The decoder uses the hidden variables generated by the encoder to reconstruct the data. These two modules are elaborated upon in detail subsequently.

### Encoder

CNNs have advantages in feature extraction, as they can automatically learn the hierarchical representation of input data, thereby eliminating the necessity for manual feature engineering. In this article, we employ a multi-scale, multi-channel CNN to adaptively extract spatial features from time series data. Each channel corresponds to a one-dimensional time series, and the feature extraction process is conducted independently for each channel, yielding local spatial features for each variable.

$$Y = f(W * K * X + B). \tag{5}$$

In this equation, $Y$ represents the feature vector obtained through the convolution operation, where $W$, $K$, and $B$ signify the weight, convolution kernel, input data, and bias value, respectively.

The computational efficiency of convolution operations achieved by CNNs through matrix multiplication enables them to automatically learn and extract valuable features from input data, greatly simplifying the feature extraction process. Consequently, CNNs can still maintain a fast training and inference speed when processing large-scale time series datasets. The convolution process applied to the time series is illustrated in Fig. 2. Furthermore, CNNs facilitate automatic feature learning and recognition across multiple scales, eliminating the need for manual design and selection of feature extractors. To capture

**Peer**J Computer Science

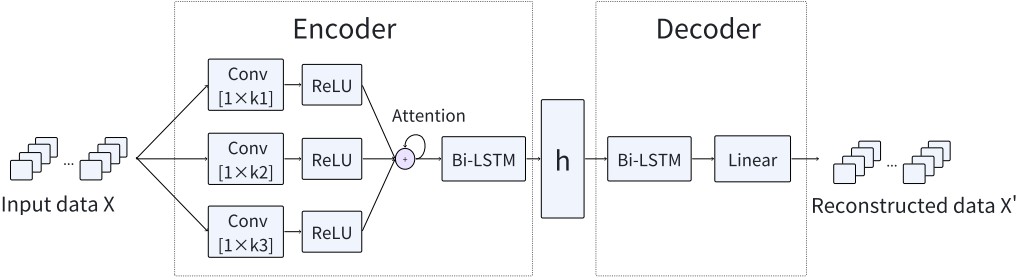

**Figure 1** Overview of the MFAM-AD model.

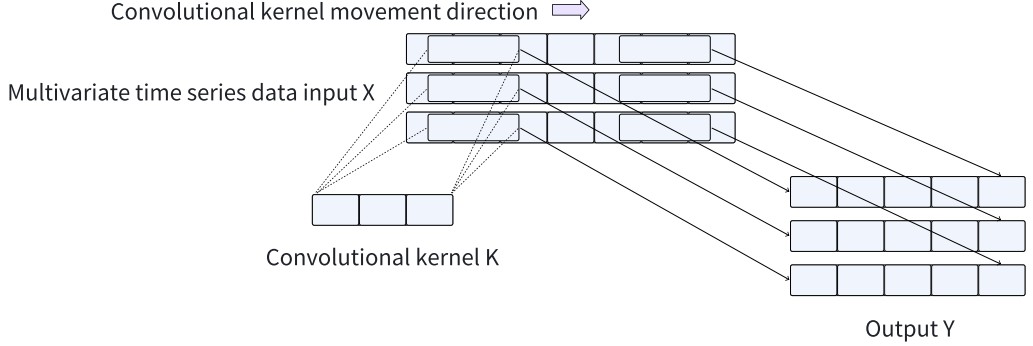

**Figure 2** Convolution process.

local spatial features across different scales of time series data, we employ CNNs with varying convolution kernel sizes, enabling adaptive and scale-specific feature extraction.

Three sets of convolutional kernels with dimensions $1 * k_i$ ($i = 1, 2, 3$) are used to extract features from time series to extract multi-scale features. These kernels are capable of extracting features from the time series data across different scales, labeled as $f_1, f_2, f_3$, and spliced together to obtain a feature F with dimensions of 3N (where N is the feature dimension).

$$F = concat([f_1, f_2, f_3]). \tag{6}$$

The attention mechanism plays a crucial role in emphasizing significant features while suppressing less relevant ones by automatically assigning weights to different features. This capability enhances the model's effectiveness in utilizing features and focusing on the essential details of the task at hand. Consequently, we fuse multi-scale feature fusion by the attention mechanism as follows:

To compute the attention score, the input matrix X is multiplied by the learnable weight matrix W, which is of the form $[3N * 1]$, resulting in the generation of attention scores (AttnScores).

$$AttnScores = F * W. \tag{7}$$

We then apply the softmax function to the attention scores to obtain the attention weights (AttnWeights).

$$AttnWeights = softmax(AttnScores). \tag{8}$$

Finally, to obtain the fused feature (Z), we perform an element-wise weighting operation using the attention weights (AttnWeights) and the original feature (X).

$$Z = F * AttnWeights. \tag{9}$$

After successfully extracting local spatial features from the time series data, the data is forwarded to the Bi-LSTM layer. This enables the model to comprehensively capture the intrinsic temporal dependencies within the data. The Bi-LSTMs are adept at simultaneously encoding sequence information in both forward and backward directions. We denote the forward and backward LSTMs as $h_f$ and $h_b$, respectively. The final hidden states of the forward and backward LSTMs, denoted by $h_f$ and $h_b$, respectively, are concatenated to form the spliced hidden variable $h$. This allows the model to fully capture sequential information in both directions, enabling more accurate temporal pattern recognition.

$$h = concat([h_f, h_b]). \tag{10}$$

## Decoder

LSTMs not only capture the temporal dynamics of data well, but also effectively solve the problems of gradient explosion and vanishing when processing large amounts of sequences. This is enabled by a distinctive internal mechanism referred to as "gates". Bi-LSTMs, which seamlessly integrate forward and backward LSTMs, provide an enhanced method for incorporating contextual information into output generation, as shown in Fig. 3. By utilizing the design principles of Bi-LSTM, our model is able to comprehensively capture the temporal features of time series data.

In the Bi-LSTM encoder, a sequence of hidden states $h = (h_1, h_2, \ldots, h_i)$ is generated given an input sequence $X = (X_1, X_2, \ldots, X_i)$. Subsequently, the Bi-LSTM decoder utilizes these hidden states $h$ to reconstruct the time series $X' = (X'_1, X'_2, \ldots, X'_i)$. During the decoding stage, the input data is reconstructed by combining the hidden states in both the forward and reverse orders.

The Bi-LSTM model outperforms traditional RNNs across multiple tasks and offers the aforementioned advantages when processing sequential data. Additionally, it handles noise and missing values in the input data remarkably well. In tasks that involve data reconstruction, the robustness of Bi-LSTMs can mitigate the impact of disturbances such as noise or missing values in the original data on the reconstruction outcomes. Consequently, the decoder component of the MFAM-AD model employs a Bi-LSTM, which is crucial for accurately reconstructing the original data using the hidden variables generated by the encoder. The primary objective of this network is to effectively reconstruct multivariate time series data by thoroughly exploiting the latent temporal features embedded in the hidden variables. By integrating the essential components of LSTMs, Bi-LSTMs emerge as a powerful deep learning model capable of effectively handling complex data with temporal

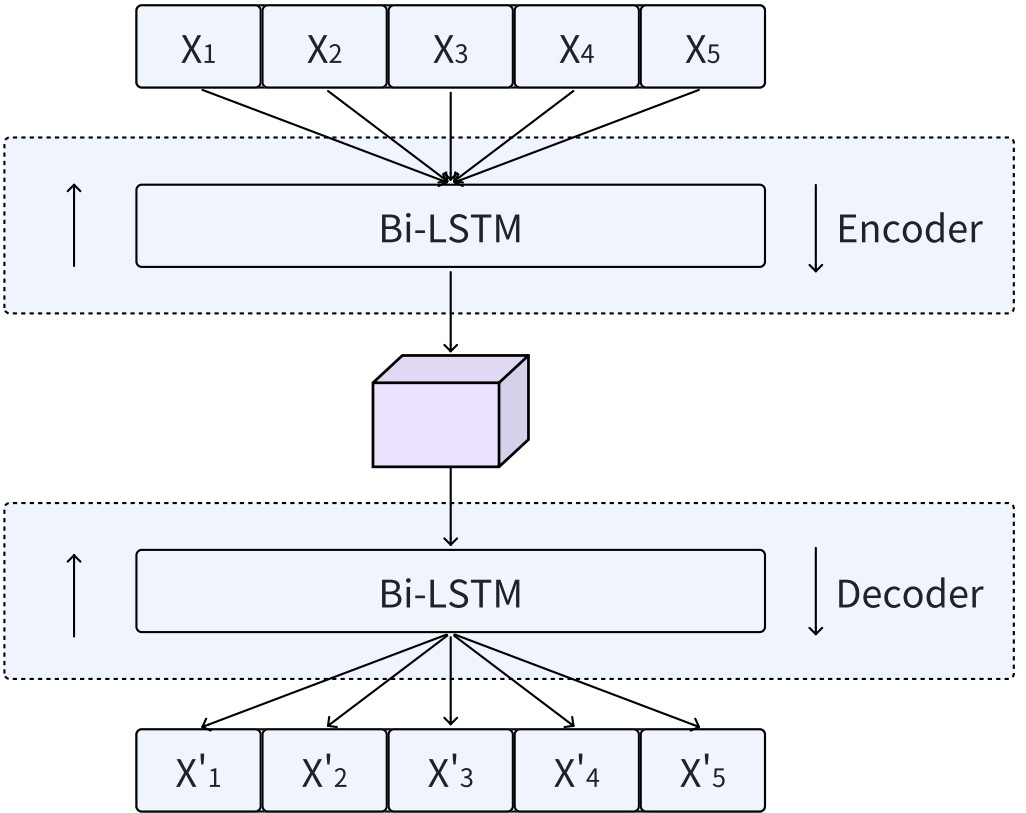

**Figure 3** **Encoding and decoding process using Bi-LSTM.**

dependencies. The Bi-LSTM decoder in the MFAM-AD model leverages its specialized memory cells and gating mechanisms to precisely capture the hidden temporal features contained within the input hidden variables from the encoder, and uses it to reconstruct the time series.

The decoder is realized through the employment of a Bi-LSTM. Input the hidden variable h containing spatiotemporal features obtained from the encoder into the Bi-LSTM to obtain data X, and output the reconstruction data X' of the original data through the fully connected network.

The entire process can be formalized using the following equations:

$$X = \text{BiLSTM}(h) \tag{11}$$

$$X' = \text{Linear}(X). \tag{12}$$

The function BiLSTM represents the operation of the bidirectional long short-term memory network, which processes the hidden variables h to produce the intermediate representation X. In the second equation, the function Linear represents the operation of the fully connected network, which transforms the intermediate representation X into the final reconstructed output X'.

## Abnormal scoring and determination

Data reconstruction serves as a crucial component in anomaly detection, aiming to minimize information loss (*Xu et al., 2018*). The term "reconstruction error" encompasses the potential loss of information incurred during the reconstruction process of input data, forming the fundamental basis of this approach. In this study, the model undergoes initial training using normal data, enabling it to capture the distinctive patterns of normal behavior. Given the relative ease with which normal data can be reconstructed compared to abnormal data, the model demonstrates high accuracy in discriminating between the two. Specifically, the reconstruction error is determined by quantifying the disparity between the input and reconstructed data.

For this purpose, a training dataset $S_N$ is curated from normal time series data, while a test dataset $T_A$ comprises anomalous time series data. The model is trained using $S_N$, with the mean square loss of the multi-step reconstruction serving as the loss function. The real data at the current time is denoted by $x_i$, the reconstructed data at the same time is represented as $r_i$, and the duration of the time window is denoted by n.

$$Loss = \sqrt{\frac{1}{n}\sum_{i=1}^{n}(x_i - r_i)^2}. \tag{13}$$

The reconstructed data in the $j$th dimension at time point $i$ is denoted as $r_i^j$, while the corresponding real data is denoted as $x_i^j$. Then, compute the average of the residuals between the actual data and the reconstructed data across all dimensions to derive the anomaly score. This score serves as the basis for identifying anomalies.

$$Score = \frac{1}{n}\sum_{j=1}^{n}|x_i^j - r_i^j|. \tag{14}$$

To determine if $x_i$ is anomaly, we compare its anomaly score with a predefined threshold r. If the score exceeds the threshold, $x_i$ is designated as anomalous; otherwise, it is considered normal. To determine the appropriate threshold r, we utilize the Peaks-Over-Threshold (POT) method, which provides a statistically robust approach for anomaly detection.

## EXPERIMENTS

### Datasets

To validate the effectiveness of MFAM-AD, we employ publicly available datasets, including MSL, SMD1-1, SMAP, NIPS-TS-SWAN, and NIPS-TS-GECCO. Table 1 provides a comprehensive overview of these datasets, outlining their respective characteristics and applications.

The MSL (Mars Science Laboratory) dataset, obtained from the Martian surface, comprises 55 dimensions per node, encompassing crucial parameters such as temperature, humidity, air pressure, and light intensity. Analysis of this dataset enables the detection of anomalies stemming from fluctuations in the Martian environment.

The SMD (Server Machine Dataset) dataset originates from an Internet company, and for our experimental analysis, we have selected the data subset marked as 1-1, referred to as

**Table 1  Description of datasets.**

| Datasets | Features | Train | Test | Anomalies |
|---|---|---|---|---|
| MSL | 55 | 58,317 | 73,729 | 10.72% |
| SMAP | 25 | 135,183 | 427,617 | 12.13% |
| SMD1-1 | 38 | 28,479 | 28,479 | 9.45% |
| NIPS-TS-SWAN | 38 | 60,000 | 60,000 | 32.6% |
| NIPS-TS-GECCO | 9 | 69,260 | 69,261 | 1.05% |

SMD1-1. The SMAP (Soil Moisture Active Passive Satellite) dataset has global soil moisture data, including information on both surface and subsurface soil moisture, measurements of soil moisture in millimeters, percentages of soil moisture profiles, and changes in both surface and subsurface soil moisture.

The NIPS-TS-SWAN dataset (*Angryk et al., 2020*) serves as a publicly accessible benchmark for multivariate time series analysis, featuring a diverse range of magnetic-field correlation parameters for active regions. Furthermore, the NIPS-TS-GECCO dataset (*Lai et al., 2021*; *Moritz et al., 2018*), a drinking water quality dataset for the "Internet of Things", was introduced at the 2018 Genetic and Evolutionary Computation Conference, offering a comprehensive view of water quality parameters.

## Experimental setup
### Baselines

To evaluate the performance of the MFAM-AD model, we have chosen nine baseline anomaly detection methods for comparison:

- PCA (*Shyu et al., 2003*): PCA identifies anomalies by projecting the data onto a low-dimensional space, preserving most of the significant features, and utilizing reconstruction error as a metric.
- KNN (*Ramaswamy, Rastogi & Shim, 2000*): KNN utilizes the distance to the nearest data point as a local density estimate to identify anomalies.
- IForest (*Liu, Ting & Zhou, 2008*): IForest achieves anomaly detection by constructing a series of segregated decision trees that treat anomalies as outliers.
- LOF (*Breunig et al., 2000*): LOF identifies outliers by comparing the local density of each data point to its neighbors.
- AE (*Hawkins et al., 2002*): AE maps the input to the hidden space and reconstructs the input. Data with large reconstruction errors will be detected as anomalies.
- EncDec-AD (*Malhotra et al., 2016*): EncDec-AD learns an intrinsic representation of the data using an LSTM-based autoencoder. Data with large reconstruction errors are considered anomalous.
- GDN (*Deng & Hooi, 2021*): GDN uses embedding vectors to learn spatial relationships between variables and utilizes an attention mechanism to make single-step predictions. Prediction errors are considered anomalous scores.
- TranAD (*Tuli, Casale & Jennings, 2022*): TranAD leverages reconstruction and prediction errors in the training phase as model loss gradients and prediction errors as anomaly scores in the testing phase.

- DCdetector (*Yang et al., 2023*): DCdetector utilizes a dual attention mechanism to extract contextual information and dependencies between variables in timing data. It employs contrastive learning to distinguish between normal and abnormal representations, enabling anomaly detection.

### Configuration

**Experimental setup:** The MFAM-AD model code is developed using PyTorch 1.13.1, a widely used deep learning framework. The training of the model is conducted on an A6000 GPU, leveraging Python 3.7 for compatibility and efficiency.

**Parameter settings:** The pertinent hyperparameter values for the model are detailed in Table 2. Specifically, the model employs the Adam optimizer with the following configurations: a learning rate of 0.001, Mean Squared Error (MSE) as the loss function, a random seed number set to 42 for reproducibility, and a time window size of 30 for appropriate temporal analysis.

### Evaluation metrics

In this study, we comprehensively evaluate the proposed model using various evaluation metrics that comprehensively assess its performance. Specifically, we consider metrics such as accuracy, precision, recall, and F1 score. In this context, TP represents the number of anomalies correctly detected by the model, FN denotes the number of anomalies missed, FP indicates the number of normal data instances incorrectly flagged as anomalies, and TN represents the number of normal data instances correctly identified. By analyzing these metrics, we aim to gain a comprehensive understanding of the model's ability to detect and classify anomalies accurately.

Accuracy, a metric that quantifies the overall detection accuracy, is computed using the following formula:

$$Accuracy = \frac{TP + TN}{TP + TN + FP + FN}. \tag{15}$$

The detection coverage is evaluated using Recall, computed with the following formula:

$$Recall = \frac{TP}{TP + FN}. \tag{16}$$

Precision assesses the detection precision and is calculated as follows:

$$Precision = \frac{TP}{TP + FP}. \tag{17}$$

F1 score synthesizes the precision and recall of the model, represented by the equation:

$$F1 = \frac{2 * (Precision * Recall)}{Precision + Recall}. \tag{18}$$

## Experimental results

Our MFAM-AD model demonstrates superior performance in comparison to various baseline models, as evident from the comparison experiments outlined in Table 3 and

**Table 2  Hyperparameter setting.**

| Hyperparameter | Interpretations | Value |
|---|---|---|
| k | Convolution kernel size | 3,5,7 |
| j | Number of convolution groups | 3 |
| c | Number of hidden layer nodes | 200 |
| n | Number of network layers | 1 |
| f | Convolutional step | 1 |

Fig. 4. Across the MSL, SMD1-1, and SMAP datasets, MFAM-AD consistently surpasses all baseline models, except for DCdetector.

Conventional anomaly detection algorithms, such as KNN and PCA, often struggle to effectively capture the diverse and intricate anomalous patterns inherent in time series data. Consequently, their performance often falls short of expectations. Similarly, the AutoEncoder model may not fully capture temporal information during time series reconstruction, thereby limiting its effectiveness and optimization potential. To address these limitations, EncDec-AD incorporates a recurrent neural network into its encoder, enabling precise temporal feature capture. However, this approach may encounter challenges when processing spatial-dimensional features.

The proposed MFAM-AD model employs a CNN network and a Bi-LSTM network to extract both spatial and temporal features, enabling the encoder to accommodate both. By utilizing parallel convolutional kernels of varying sizes, our model can capture intricate patterns in time series data, thereby enhancing anomaly detection capabilities on short sequences. Additionally, through the utilization of Bi-LSTM, the model effectively incorporates both past and future contextual information, thereby more accurately modeling dependencies within sequences. The efficacy of the model is further underscored by its ability to extract both temporal and local spatial features from the time series data, resulting in significantly higher F1 scores on three datasets compared to the baseline model and common machine learning methods. However, our model slightly underperforms compared to the recently proposed DCdetector model. This can be attributed to DCdetector's ability to directly leverage original data for comparative learning, thus mitigating potential information loss associated with data reconstruction.

We compare our model against the deep learning baseline using various evaluation metrics, including the Matthews correlation coefficient (MCC), affiliation precision/recall pair (Aff-P/Aff-R) (*Huet, Navarro & Rossi, 2022*), and volume under the surface (VUS) (*Paparrizos et al., 2022*). The commonly used F1 score fails to capture the impact of anomalous events, whereas Aff-P and Aff-R consider the temporal differences between actual and predicted events. Additionally, the VUS metric incorporates the ROC curve, providing a comprehensive evaluation of the trade-offs associated with abnormal events. These advanced evaluation metrics offer a more thorough and precise perspective on model performance. As shown in Table 4 and Fig. 5, our model achieves significantly higher Aff-P

**Table 3  Comparison of MFAM-AD performance with each baseline model on MSL, SMD1-1, and SMAP datasets (the best results are shown in bold and the next best results are shown in underline).**

| Model | MSL | | | | SMD1-1 | | | | SMAP | | | |
|---|---|---|---|---|---|---|---|---|---|---|---|---|
| | Acc | Rec | Pre | F1 | Acc | Rec | Pre | F1 | Acc | Rec | Pre | F1 |
| KNN | 0.8572 | 0.9654 | 0.4222 | 0.5875 | 0.5236 | 0.9999 | 0.1657 | 0.2842 | 0.7239 | 0.9585 | 0.3117 | 0.4704 |
| PCA | 0.9054 | 0.9687 | 0.5278 | 0.6833 | 0.4977 | 0.9999 | 0.1585 | 0.2736 | 0.8824 | 0.5907 | 0.5367 | 0.5624 |
| IForest | 0.9158 | 0.9687 | 0.5577 | 0.7078 | 0.6014 | 0.9999 | 0.1918 | 0.3219 | 0.8800 | 0.5907 | 0.5278 | 0.5575 |
| LOF | 0.8289 | 0.9525 | 0.3766 | 0.5398 | 0.4911 | 0.9999 | 0.1568 | 0.2710 | 0.6925 | 0.9633 | 0.2892 | 0.4449 |
| AE | 0.8985 | 0.0428 | 0.8737 | 0.0815 | 0.9724 | 0.9999 | 0.7741 | 0.8727 | 0.9240 | 0.4400 | 0.9282 | 0.5969 |
| EncDec-AD | 0.9071 | 0.1253 | 0.9456 | 0.2213 | 0.9527 | 0.9999 | 0.6669 | 0.8002 | 0.9169 | 0.3956 | 0.8984 | 0.5494 |
| GDN | 0.9525 | 0.5714 | **0.9628** | 0.7172 | 0.8462 | 0.9999 | 0.3811 | 0.5518 | 0.9409 | 0.5495 | **0.9804** | 0.7042 |
| TranAD | 0.8032 | 0.7878 | 0.3213 | 0.4564 | 0.9730 | 0.9988 | 0.7771 | 0.8741 | 0.8797 | 0.6401 | 0.5265 | 0.5778 |
| DCdetector | 0.9812 | **0.9721** | 0.9209 | **0.9458** | **0.9946** | 0.9981 | **0.9508** | **0.9739** | **0.9916** | **0.9864** | 0.9492 | **0.9674** |
| MFAM-AD | **0.9821** | 0.9556 | 0.8839 | 0.9184 | 0.9840 | **0.9999** | 0.8555 | 0.9221 | 0.9807 | 0.8853 | 0.9608 | 0.9215 |

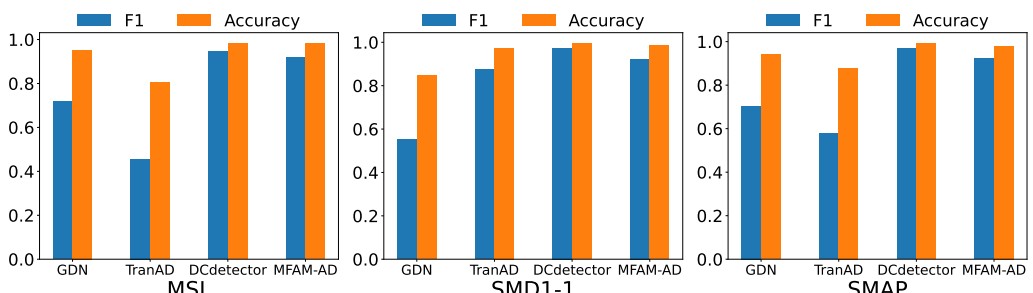

**Figure 4  Comparison of F1 and accuracy of MFAM-AD with GDN, TranAD, and DCdetector.**

scores compared to the models in the baseline, indicating its superior performance across the three datasets.

To comprehensively evaluate the performance of our model, we conducted comparative experiments with the DCdetector model using the NIPS-TS-SWAN and NIPS-TS-GECCO datasets. The characteristic of these two datasets is that NIPS-TS-SWAN has the highest percentage of anomalies (32.6%), while NIPS-TS-GECCO has the lowest percentage of anomalies (1.05%). As shown in Table 5 and Fig. 6, our model performs more comprehensively on both datasets than DCdetector, suggesting that it is more applicable to difficult anomaly detection tasks.

## Ablation studies

The MFAM-AD model encompasses two primary components: the Attention Mechanism Feature Fusion module (AMFF) and the Multi-scale Feature Extraction module (MFE). To assess the individual contributions of each component to the overall model performance, three distinct networks were devised in this ablation study: the Base network, the enhanced Base+MFE network, and the further augmented Base+MFE+AMFF network.

- Base: It represents the baseline model, utilizing a single convolutional kernel for feature extraction.

**Table 4** Performance of MFAM-AD vs. deep learning models in the baseline model on MSL, SMD1-1, and SMAP datasets using the new metrics (the best results are shown in bold and the next best results are shown in underline).

| DataSet | Method | MCC | Aff-P | Aff-R | R_A_R | R_A_P | V_ROC | V_PR |
|---------|--------|-----|-------|-------|-------|-------|-------|------|
| MSL | AE | 0.1588 | 0.5298 | **0.9791** | 0.7642 | 0.3396 | 0.7694 | 0.3433 |
| | EncDec-AD | 0.1709 | 0.4761 | 0.3663 | 0.5153 | 0.1230 | 0.5196 | 0.1245 |
| | GDN | 0.1318 | 0.5095 | 0.3030 | 0.5527 | 0.2705 | 0.5568 | 0.2632 |
| | TranAD | 0.6781 | 0.6322 | 0.1993 | 0.6133 | 0.6404 | 0.6081 | 0.6279 |
| | DCdetector | **0.9394** | 0.5117 | 0.9698 | **0.9119** | **0.8896** | **0.8964** | **0.8762** |
| | MFAM-AD | 0.8984 | **0.9070** | 0.0593 | 0.6416 | 0.6764 | 0.6272 | 0.6611 |
| SMD1-1 | AE | 0.5517 | 0.6352 | 0.6692 | 0.6314 | 0.6771 | 0.6428 | 0.6672 |
| | EncDec-AD | 0.3336 | 0.6199 | 0.5076 | 0.6882 | 0.6923 | 0.6774 | 0.6472 |
| | GDN | 0.6360 | 0.6328 | 0.6621 | 0.6557 | 0.7330 | 0.6700 | 0.7337 |
| | TranAD | 0.3439 | 0.6365 | 0.5299 | 0.7299 | 0.6823 | 0.7113 | 0.6377 |
| | DCdetector | **0.9713** | 0.5297 | **0.9252** | **0.9480** | **0.9326** | **0.9154** | **0.9036** |
| | MFAM-AD | 0.9225 | **1.0** | 0.0999 | 0.5974 | 0.6662 | 0.5977 | 0.6654 |
| SMAP | AE | −0.0018 | 0.6304 | 0.1224 | 0.4959 | 0.1271 | 0.4869 | 0.1111 |
| | EncDec-AD | −0.0021 | 0.5966 | 0.1177 | 0.5141 | 0.1701 | 0.5046 | 0.1515 |
| | GDN | 0.0055 | 0.5594 | 0.1573 | 0.4778 | 0.1056 | 0.4765 | 0.1015 |
| | TranAD | 0.7084 | 0.6172 | 0.4261 | 0.5436 | 0.4616 | 0.5485 | 0.4644 |
| | DCdetector | **0.9628** | 0.5004 | **0.9715** | **0.9509** | **0.9358** | **0.9356** | **0.9225** |
| | MFAM-AD | 0.9116 | **0.9038** | 0.0399 | 0.5900 | 0.6193 | 0.5767 | 0.6041 |

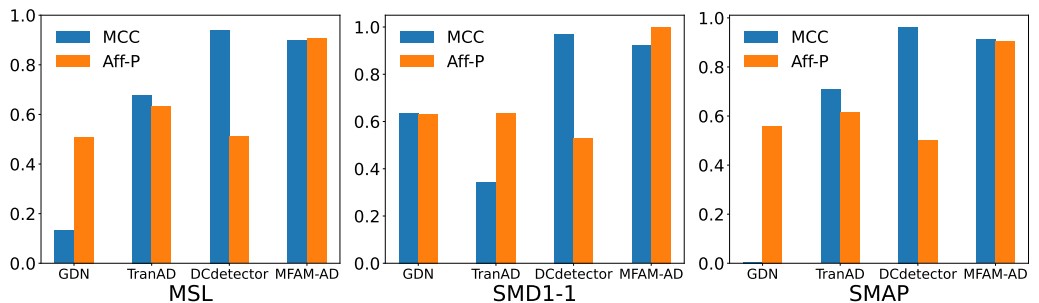

**Figure 5** Comparison of MCC and Aff-P of MFAM-AD with GDN, TranAD and DCdetector.

- Base+MFE: It extends the Base model by incorporating the Multi-scale Feature Extraction module, which employs three distinct convolutional kernels of varying sizes to extract features. The extracted features from different scales are then fused through a straightforward splicing process.
- Base+MFE+AMFF: Building upon the Base+MFE model, this enhanced variant introduces an attention mechanism for feature fusion, thereby augmenting the fusion process with attention-based weighting, further optimizing the integration of multi-scale features.

As shown in Fig. 7, the experimental results suggest that the adoption of a multi-scale feature fusion strategy may inadvertently introduce noise and redundant information,

slightly compromising the model's effectiveness. However, the subsequent integration of the attention mechanism markedly boosted the model's performance. The combination of the attention mechanism with multi-scale feature fusion effectively mitigates the model's sensitivity to noise and redundant information, resulting in a substantial improvement in its performance in anomaly detection tasks. The attention mechanism endows the model with the ability to dynamically adjust its focus on features across different scales based on the input data, thereby optimizing its capability to detect anomalies.

### Hyperparameter sensitivity

We conducted a series of rigorous experiments to validate several crucial hyperparameters, including the number of encoding layers (It is the hyperparameter n mentioned earlier, hereinafter referred to as Encode-Layer Number) and the number of hidden layer nodes (It is the hyperparameter c mentioned earlier, hereinafter referred to as Hidden-Size).

The number of hidden layer nodes Hidden-Size is an important parameter. As a hyperparameter of the hidden channels, it may have impacts on model performance, memory cost, and running efficiency. We set Hidden-Size $\in \{50, 100, 150, 200\}$ as suggested hyperparameters. For all datasets, the performance of the model will improve as the Hidden-Size increases, but in order to consider the inference efficiency of the model, we only set the Hidden-Size to be the largest value among the experimental values.

Many deep models' performances are dependent on the number of network layers Encode-Layer Number. We set Encode-Layer Number $\in \{1, 2, 3, 4\}$ as suggested hyperparameters. Our model can gain the best performance in no more than 3 layers and will not fail with too few encoder layers or overfit with too many encoder layers.

As shown in Fig. 8, the experimental outcomes revealed that these two parameters exert a significant influence on the overall performance of the model. Following tests and a comprehensive analysis of the model's inference efficiency, it was ultimately determined that an embedding dimension of 200 offers an optimal balance between maximizing inference speed and maintaining model performance. Based on these findings, the optimal number of coding layers was also experimentally confirmed; specifically, the model achieved the best performance when the number of coding layers was set to 1.

## CONCLUSION AND FUTURE WORKS

In this article, we introduce MFAM-AD, a state-of-the-art multivariate time series anomaly detection model that uses the advanced architecture of deep learning. MFAM-AD seamlessly integrates the powerful feature extraction capabilities of convolutional neural networks (CNNs) with the temporal modeling strengths of bidirectional long short-term memory networks (Bi-LSTMs). By leveraging time series reconstruction, the model efficiently identifies and captures subtle anomaly patterns. Furthermore, we enhance the model's overall performance and focus on crucial features by introducing an attention fusion mechanism. This innovative component allows the model to maintain excellent anomaly detection capabilities even in noisy data environments. Extensive experimental results demonstrate that MFAM-AD outperforms the reconstructed model in the baseline on the MSL, SMD1-1, and SMAP datasets. Moreover, it surpasses the DCdetector model on the

Xia et al. (2024), *PeerJ Comput. Sci.*, DOI 10.7717/peerj-cs.2201

**Table 5** **Performance comparison between MFAM-AD and DCdetector on NIPS-TS-SWAN and NIPS-TS-GECCO datasets (we took a few key metrics, the best results are shown in bold and the next best results are shown in underline).**

| DataSet | Method | Acc | Rec | Pre | F1 | MCC | Aff-P | Aff-R | R_A_R | R_A_P | V_ROC | V_PR |
|---|---|---|---|---|---|---|---|---|---|---|---|---|
| NIPS-TS-SWAN | DCdetector | 0.8601 | 0.5890 | 0.9678 | 0.7323 | 0.6815 | 0.5040 | 0.0386 | 0.8801 | 0.9481 | 0.8604 | 0.9354 |
| | MFAM-AD | 0.8760 | 0.6660 | 0.9352 | **0.7780** | **0.7191** | **0.7798** | **0.7270** | 0.7830 | 0.8093 | 0.7806 | 0.8011 |
| NIPS-TS-GECCO | DCdetector | 0.9851 | 0.5178 | 0.3569 | 0.4226 | 0.4227 | 0.5199 | **0.9041** | 0.6086 | 0.3100 | 0.6045 | 0.3054 |
| | MFAM-AD | 0.9929 | 0.3520 | 0.9448 | **0.5129** | **0.5893** | **0.7908** | 0.6661 | 0.7753 | 0.4686 | 0.7306 | 0.4224 |

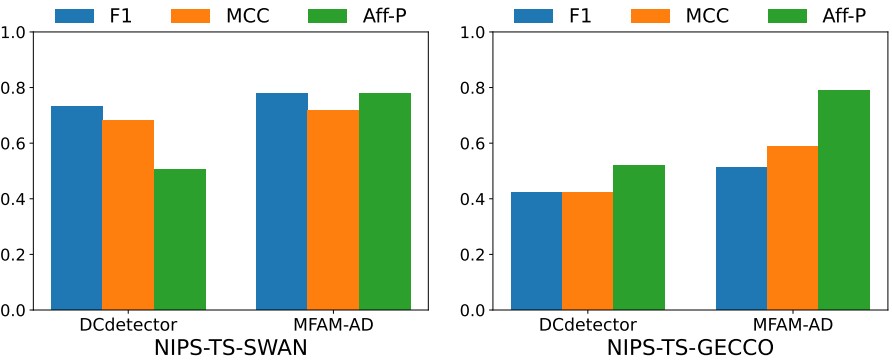

**Figure 6** Comparison of F1, MCC, and Aff-P of MFAM-AD and DCdetector.

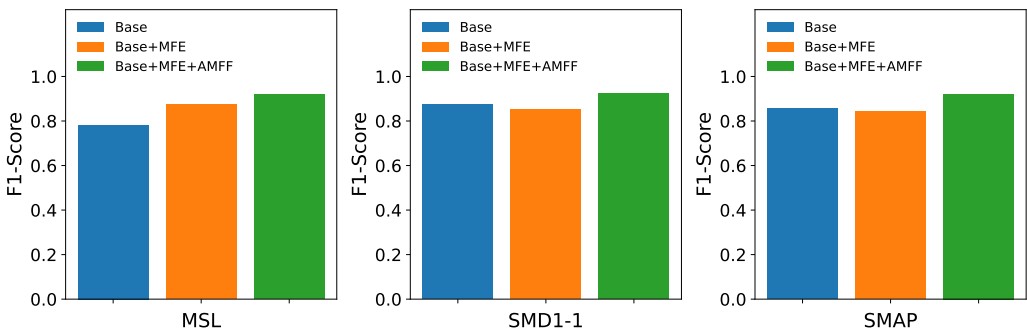

**Figure 7** Results of ablation studies on public datasets MSL, SMD1-1, SMAP.

more challenging NIPS-TS-SWAN and NIPS-TS-GECCO datasets. These findings validate the MFAM-AD model's efficacy in multivariate time series anomaly detection, highlighting its potential in various real-world applications.

Our research establishes a solid foundation for leveraging multi-scale features to optimize unsupervised anomaly detection. However, there are still some limitations inherent in this work: MFAM-AD is unable to analyze time series with dependencies due to the lack of integration of time series dependencies into the modeling process. Additionally, MFAM-AD may not perform well on low-quality datasets since it is trained entirely on normal data, which requires high quality.

Given the limitations in this study, the further work can be carried out in the following directions in the future: Firstly, incorporating dependencies between time series into the modelling process. While the current approach leverages CNNs and Bi-LSTMs to capture spatial and temporal features, it overlooks inherent dependencies among time series. Future work can involve incorporating models like graph attention networks into MFAM-AD, enabling it to model dependencies between time series effectively. Secondly, exploring more advanced feature fusion techniques offers promising avenues for improving the model's ability to filter out irrelevant information. While the current approach leverages the attention mechanism for feature fusion, there are likely other innovative methods or

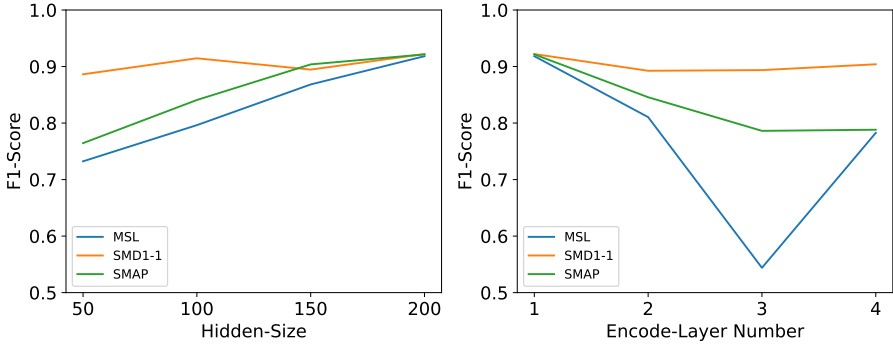

**Figure 8** Parameter sensitivity studies of main hyperparameters in MFAM-AD.

combinations of techniques that could yield superior results. It is necessary to explore novel fusion strategies or combinations of existing techniques to further boost detection performance. Thirdly, addressing the challenges associated with enhancing data quality remains a crucial focus for this area. While this work achieved excellent results on the datasets in this article, it is crucial to consider the potential presence of noise when applying MFAM-AD to other datasets. Future algorithms can incorporate data quality control components to design innovative solutions that effectively improve data quality.

In general, MFAM-AD provides significant insights on improving the accuracy of multivariate time series anomaly detection models. We are hopeful that our work will serve as a valuable contribution to the field of anomaly detection, sparking further advancements in these critical areas.

### Funding
This work was funded by the National Natural Science Foundation of China (No. 61562010 and 71964009) and the Research Projects of the Science and Technology Plan of Guizhou Province (No. [2021] 449, [2021] 261, [2023] 010, [2023] 276, [2023] 338). The funders had no role in study design, data collection and analysis, decision to publish, or preparation of the manuscript.

### Grant Disclosures
The following grant information was disclosed by the authors:
The National Natural Science Foundation of China: 61562010, 71964009.
The Research Projects of the Science and Technology Plan of Guizhou Province: (No. [2021] 449, [2021] 261, [2023] 010, [2023] 276, [2023] 338).

### Competing Interests
The authors declare there are no competing interests.

## Author Contributions

- Shengjie Xia conceived and designed the experiments, performed the experiments, analyzed the data, performed the computation work, prepared figures and/or tables, authored or reviewed drafts of the article, and approved the final draft.
- Wu Sun conceived and designed the experiments, prepared figures and/or tables, and approved the final draft.
- Xiaofeng Zou conceived and designed the experiments, prepared figures and/or tables, and approved the final draft.
- Panfeng Chen analyzed the data, authored or reviewed drafts of the article, and approved the final draft.
- Dan Ma performed the experiments, prepared figures and/or tables, and approved the final draft.
- Huarong Xu performed the experiments, authored or reviewed drafts of the article, and approved the final draft.
- Mei Chen analyzed the data, authored or reviewed drafts of the article, and approved the final draft.
- Hui Li analyzed the data, authored or reviewed drafts of the article, and approved the final draft.

## Data Availability

The code and datasets used for all the experiments are available in the Supplemental Files.

The SMD dataset is available at GitHub: https://github.com/NetManAIOps/OmniAnomaly/tree/master/ServerMachineDataset.

The MSL/SMAP dataset is available at GitHub and Zenodo:

- https://github.com/khundman/telemanom

- xia, . shengjie . (2024). The dataset used for the MFAM-AD model [Data set]. Zenodo. https://doi.org/10.5281/zenodo.12769004,

The MSL/SMAP data label is available at:

https://raw.githubusercontent.com/khundman/telemanom/master/labeled_anomalies.csv.

The NIPS-TS-SWAN dataset is available at Harvard Dataverse: Angryk, Rafal; Martens, Petrus; Aydin, Berkay; Kempton, Dustin; Mahajan, Sushant; Basodi, Sunitha; Ahmadzadeh, Azim; Xumin Cai; Filali Boubrahimi, Soukaina; Hamdi, Shah Muhammad; Schuh, Micheal; Georgoulis, Manolis, 2020, "SWAN-SF", https://doi.org/10.7910/DVN/EBCFKM, Harvard Dataverse, V1. We used the DCdetector processed format.

The DCdetector source code is available at GitHub: https://github.com/DAMO-DI-ML/KDD2023-DCdetector.

The NIPS-TS-GECCO dataset is available at Zenodo: Moritz, S., Rehbach, F., Chandrasekaran, S., Rebolledo, M., & Thomas Bartz-Beielstein. (2018). GECCO Industrial Challenge 2018 Dataset: A water quality dataset for the 'Internet of Things: Online Anomaly Detection for Drinking Water Quality' competition at the Genetic

and Evolutionary Computation Conference 2018, Kyoto, Japan. [Data set]. The Genetic and Evolutionary Computation Conference (GECCO), Kyoto, Japan. Zenodo. https://doi.org/10.5281/zenodo.3884398. We used the same DCdetector processed format.

All datasets are available at Zenodo: xia, . shengjie . (2024). The dataset used for the MFAM-AD model [Data set]. Zenodo. https://doi.org/10.5281/zenodo.11392850.

## Supplemental Information

Supplemental information for this article can be found online at http://dx.doi.org/10.7717/peerj-cs.2201#supplemental-information.

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
