# Peer review of "MFAM-AD: an anomaly detection model for multivariate time series using attention mechanism to fuse multi-scale features"

_PeerJ Computer Science, doi:10.7717/peerj-cs.2201_

## Round 0.1 · original submission · Minor Revisions

Please revise the paper according to the comments. Then it will be evaluated again.

Reviewer 1 ·

Basic reporting

This paper proposed an anomaly detection model for multivariate time series using an attention mechanism to fuse multi-scale features. This Topic is very important and interesting. It fails to meet the journal standards.

Experimental design

This paper conducted experiments on two challenging datasets, the experimental results show its superiority. It fails to meet the journal standards.

Validity of the findings

This paper proposed a cutting-edge model for multivariate time series anomaly detection.

Additional comments

1. It is recommended to redraw the figures to match the font of the paper.
2. There are grammar errors in the paper, and writing can be improved.
3. In the abstract, the paper claimed that "Specifically, our MFAM-AD model achieves F1 scores that exceed the current state-of-the-art DCdetector model by 5.9% and 17.6%, respectively.", the performance improvement is certainly good, but a 17.6% increase feels a bit too high, it would be best to fully explain.

Reviewer 2 ·

Basic reporting

Methodology of MFAM-AD is both reasonable and robust, as evidenced by comprehensive experimental validation that confirms its superior performance. However, minor revisions to correct typos and language issues would enhance the paper, and a clearer discussion of its limitations would be beneficial.

Experimental design

The concept underlying the MFAM-AD model is insightful, as it combines the strengths of CNNs and Bi-LSTM to seamlessly integrate temporal dependencies and multi-scale spatial features, enhancing anomaly detection. The experimental design is comprehensive, ensuring robust and convincing evaluation.

Validity of the findings

This paper provides sufficient detail to enable replication of the study. The experimental validation is thorough, conducted on two challenging datasets, and the results demonstrate the superior performance of the MFAM-AD model in multivariate time series anomaly detection.

Reviewer 3 ·

Basic reporting

The manuscript provides sufficient background and context about the MFAM-AD model and its application in multivariate time series anomaly detection. The research question is well-defined, relevant, and meaningful. The structure of the article is reasonable, with well-organized sections, figures, and tables. The paper is self-contained, presenting relevant results to the hypothesis. However, there are a few areas that still need to be improved to increase the fluency and quality of writing.

Experimental design

The methods, including the architecture of the MFAM-AD model and the experimental validation process, are described with sufficient detail and information to replicate. The evaluation is rigorous, performed to a relatively high technical standard. While the methodology section provides sufficient detail, additional information regarding the training process and how to make hyperparameter selection may increase the reproducibility of the study.

Validity of the findings

The underlying data seems robust, statistically sound, and well-controlled. However, in the experimental results section, some discussions about why the EDD model outperforms other models lack adequate explanation. The conclusions are clearly articulated and closely tied to the original research question.

---

## Round 0.2 · accepted · Accept

The paper has been revised well. It can be accepted currently.

Reviewer 1 ·

Basic reporting

Clear and unambiguous, professional English used throughout. Literature references, sufficient field background/context provided. Professional article structure, figures, tables. Raw data shared. The manuscript has been revised according to the reviewers' suggestions.

Experimental design

Research question well defined, relevant & meaningful. It is stated how research fills an identified knowledge gap.

Validity of the findings

All underlying data have been provided; they are robust, statistically sound, & controlled.

Additional comments

The manuscript has been revised according to the reviewers' suggestions.

Reviewer 2 ·

Basic reporting

Good.

Experimental design

Professional.

Validity of the findings

Meaningful.

Additional comments

I have no further concerns. I suggest accepting this paper.

Reviewer 3 ·

Basic reporting

no comment

Experimental design

no comment

Validity of the findings

no comment

Additional comments

The author has made revisions to the paper based on my previous review comments, improving its quality.